# Low Dose Low-Molecular-Weight Heparin for Thrombosis Prophylaxis: Systematic Review with Meta-Analysis and Trial Sequential Analysis

**DOI:** 10.3390/jcm8122039

**Published:** 2019-11-21

**Authors:** Ruben J. Eck, Wouter Bult, Jørn Wetterslev, Reinold O. B. Gans, Karina Meijer, Iwan C. C. van der Horst, Frederik Keus

**Affiliations:** 1Department of Internal Medicine, University Medical Center Groningen, 9713 GZ Groningen, The Netherlands; r.o.b.gans@umcg.nl; 2Department of Clinical Pharmacy and Pharmacology, University Medical Center Groningen, 9713 GZ Groningen, The Netherlands; w.bult@umcg.nl; 3The Copenhagen Trial Unit, Centre for Clinical Intervention Research, Copenhagen University Hospital-Rigshospitalet, DK-2100 Copenhagen, Denmark; joern.wetterslev@ctu.dk; 4Department of Hematology, University Medical Center Groningen, 9713 GZ Groningen, The Netherlands; k.meijer@umcg.nl; 5Department of Critical Care, University Medical Center Groningen, 9713 GZ Groningen, The Netherlandsf.keus@umcg.nl (F.K.); 6Department of Intensive Care, Maastricht University Medical Center, 6229 HX Maastricht, The Netherlands

**Keywords:** low-molecular-weight heparin, venous thromboembolism, meta-analysis

## Abstract

International guidelines recommend low-molecular-weight heparin (LMWH) as first-line pharmacological option for the prevention of venous thromboembolism (VTE) in many patient categories. Guidance on the optimal prophylactic dose is lacking. We conducted a systematic review with meta-analysis and trial sequential analysis (TSA) of randomized controlled trials to assess benefits and harms of low-dose LMWH versus placebo or no treatment for thrombosis prophylaxis in patients at risk of VTE. PubMed, Cochrane Library, Web of Science, and Embase were searched up to June 2019. Results were presented as relative risk (RR) with conventional and TSA-adjusted confidence intervals (CI). Forty-four trials with a total of 22,579 participants were included. Six (14%) had overall low risk of bias. Low-dose LMWH was not statistically significantly associated with all-cause mortality (RR 0.99; 95%CI 0.85–1.14; TSA-adjusted CI 0.89–1.16) but did reduce symptomatic VTE (RR 0.62; 95%CI 0.48–0.81; TSA-adjusted CI 0.44–0.89) and any VTE (RR 0.61; 95%CI 0.50–0.75; TSA-adjusted CI 0.49–0.82). Analyses on major bleeding (RR 1.07; 95%CI 0.72–1.59), as well as serious adverse events (SAE) and clinically relevant non-major bleeding were inconclusive. There was very low to moderate-quality evidence that low-dose LMWH for thrombosis prophylaxis did not decrease all-cause mortality but reduced the incidence of symptomatic and asymptomatic VTE, while the analysis of the effects on bleeding and adverse events remained inconclusive.

## 1. Introduction

Venous thromboembolism (VTE) is a common cause of morbidity and mortality in hospitalized and non-hospitalized patients [1]. The American Society of Hematology and American College of Chest Physicians guidelines recommend low-molecular-weight heparin (LMWH) as a first-line pharmacological option for most patients at risk of VTE [2,3]. Several prophylactic doses and types of LMWH are used worldwide, which is reflected by differences in national summaries of product characteristics (SPCs) and dosing regimens of randomized controlled trials (RCTs). There is no high-quality evidence or guidance on the optimal prophylactic LMWH dose. Preceding systematic reviews on thrombosis prophylaxis have not specifically assessed benefits and harms associated with different LMWH doses [4,5,6,7,8,9,10]. In addition, there have been very few direct comparisons of prophylactic LMWH dose regimens, and therefore indirect evidence could provide a ‘second best’ estimate of benefits and harms. 

There is no generally accepted definition of different prophylactic LMWH dose categories, which is why we previously categorized LMWH thrombosis prophylaxis regimens as either ‘low-dose’ or ‘intermediate-dose’, based on different registered doses in SPCs worldwide [11]. Using this approach in a previous meta-analysis, we found that intermediate-dose LMWH, compared with placebo or no treatment, was associated with a significant decrease in symptomatic VTE, at the cost of an increase in major bleeding [11]. The main objective of the current study was to perform a systematic review with meta-analysis and trial sequential analysis (TSA) comparing benefits and harms of low-dose LMWH versus placebo or no treatment for thrombosis prophylaxis in all types of patients at risk of VTE [12].

## 2. Materials and Methods

We conducted this systematic review according to a pre-published protocol on PROSPERO (https://www.crd.york.ac.uk/prospero/display_record.php?ID=CRD42019124722) following the methodology suggested by Jakobsen et al, the Cochrane Handbook for Systematic Reviews of Interventions, the PRISMA (Preferred Reporting Items for Systematic Reviews and Meta-Analyses) statement, and the GRADE (Grading of Recommendations Assessment, Development, and Evaluation) recommendations [12,13,14,15].

### 2.1. Study Selection

#### 2.1.1. Patients

Studies were considered for inclusion irrespective of language, blinding, publication status, or sample size. We included RCTs with adult patients allocated to receive thrombosis prophylaxis using either low-dose LMWH, placebo, or no treatment, regardless of their underlying disease or whether they were admitted to the hospital or visited the outpatient clinic. 

#### 2.1.2. Interventions

The experimental intervention was low-dose LMWH, irrespective of LMWH type or duration of treatment. We a priori defined ‘low dose’ in our protocol according to the SPCs as approved by the US Food and Drug Administration, the European Medicines Agency, and several national authorities (Table 1). If different LMWHs or (weight-adjusted) doses were used in one trial, we classified the dose according to what was used most frequently. We included trials evaluating ultra-low-molecular-weight heparins and LMWHs not listed in Table 1 (e.g., LMWHs we were unable to classify into a specific dose) in a sensitivity analysis. The control intervention was placebo or no treatment. Co-interventions such as mechanical compression devices were allowed if they were applied in both treatment groups.

#### 2.1.3. Outcomes

Predefined co-primary outcomes were all-cause mortality, symptomatic VTE, and major bleeding. Secondary outcomes were serious adverse events (SAE), clinically relevant non-major bleeding, and any VTE (including both symptomatic and asymptomatic events). All outcomes were assessed at maximum follow-up. VTE was defined as deep vein thrombosis or pulmonary embolism, and the diagnosis was accepted when objectified by an imaging technique or autopsy. We made no distinction between distal or proximal, or lower versus upper extremity thrombosis. Major bleeding and clinically relevant non-major bleeding were defined according to trial criteria. SAE were defined according to the International Conference on Harmonisation of Good Clinical Practice definitions (ICH-GCP) [16].

### 2.2. Data Sources and Searches

We searched the Cochrane Central Register of Controlled Trials (CENTRAL) in The Cochrane Library, PubMed/MEDLINE, EMBASE and Web of Science (Appendix A). References of identified studies were screened to identify further relevant trials. Finally, we searched the World Health Organization’s International Clinical Trials Registry and ClinicalTrials.gov for ongoing trials (Appendix A). The search was last updated on 10 June 2019. 

### 2.3. Data Extraction and Quality Assessment

Two authors (RJE, WB) independently identified trials for inclusion. Trials excluded on the basis of full text were listed with reasons for exclusion. We extracted information on characteristics (year of publication, country, numbers of sites and patients enrolled), participants (age, sex, eligibility criteria), interventions (type, dose, and duration of LMWH treatment), and outcomes. We resolved differences in opinion through discussion. Two authors (R.J.E., W.B.) independently assessed risks of bias of the included trials according to the revised Cochrane risk of bias tool version 2 [17] in the following five domains: “Bias arising from the randomization process’’, “Bias due to deviations from intended interventions’’, “Bias due to missing outcome data’’, “Bias in measurement of the outcome’’, “Bias in selection of the reported result’’. RCTs were classified as ‘overall low risk of bias’ when all bias domains were judged as ‘low risk’. Conversely, trials were classified as ‘overall high risk of bias’ when ‘some concerns’ or ‘high risk’ was judged in one or more domains [18]. Publication bias was assessed by inspecting funnel plots for signs of asymmetry when 10 or more trials were included in the analyses [12,14]. 

### 2.4. Data Synthesis and Analysis

We calculated relative risk (RR) with both conventional 95% confidence intervals (CIs) and TSA-adjusted CI if there were two or more trials for each outcome.

#### 2.4.1. Assessment of Significance

We used adjusted thresholds for statistical significance to correct for multiplicity issues due to repeated testing. An alpha of 0.025 was used for the co-primary and secondary outcomes to keep the family-wise error rate at a maximum of 5% [14]. In case of statistically significant RR, we calculated numbers needed to treat (NNT) or numbers needed to harm (NNH) with 97.5% CI.

#### 2.4.2. Meta-Analysis

Data were pooled using both a fixed-effect and a random-effects model. In case of discrepancy between the models, we emphasized the most conservative estimate. Analyses were performed on an intention-to-treat basis whenever possible or otherwise using an ‘available-case analysis’. 

#### 2.4.3. Trial Sequential Analysis

Conventional meta-analyses may result in type-I errors due to risks of random error when few data have been collected or due to repeated significance testing when a meta-analysis is updated with new trials [19,20,21,22,23]. TSA is a sequential meta-analysis method that combines required information size estimation (i.e., the number of patients needed to detect an a priori specified relative risk reduction) with an adjusted threshold for statistical significance [21,22]. This adjusted threshold is more conservative when data are sparse and becomes progressively more lenient as the accumulated sample size approaches the required information size. Accordingly, the TSA-adjusted CI is initially wider than the conventional 95% CI, but when the required information size has been reached, they become identical. The required information size is calculated on the basis of the unweighted event proportion in the control group, the assumption of a plausible relative risk reduction/increase (RRR/RRI), and the anticipated heterogeneity variance (D^2^) of the meta-analysis. We applied TSA to all outcomes, using the control event proportion from the actual meta-analyses; D^2^ as suggested by the meta-analysis; alpha of 2.5%; beta of 90%; and an anticipated RRR/RRI of 20%. 

#### 2.4.4. Assessment of Heterogeneity 

Statistical heterogeneity I^2^ was explored by the chi-squared test with significance set at a *p*-value of 0.10. The quantity of heterogeneity was also measured by D^2^ [24]. Clinical heterogeneity was explored by conducting explorative subgroup analyses. 

#### 2.4.5. Subgroup Analysis

We performed subgroup analyses according to overall risk of bias (low vs. high), type of patients, LMWH type, duration of the intervention (less vs. more than 30 days), and length of follow-up (less vs. more than 30 days). Statistically significant subgroup differences (test of interaction *p* < 0.05) provided evidence of an intervention effect pending the subgroup.

#### 2.4.6. Sensitivity Analysis

All analyses were re-conducted including trials that evaluated LMWH types not covered by Table 1. In addition, sensitivity TSAs were conducted using an RRR as suggested by the overall low-risk-of-bias studies and using a D^2^ of 25% if the actual D^2^ was 0%. In case of rare events (<2% in the control group), TSA was also performed using Peto’s odds ratio. 

SAE are often inconsistently reported and, in addition to assessing SAE according to trial reporting, we estimated the number of patients with one or more SAE using two methods: (1) the highest proportion of either reported mortality, symptomatic VTE, or major bleeding in each trial and (2) all mortality, SAE, symptomatic VTE, and major bleeding events cumulated in each trial. The idea is that the ‘true proportion’ of SAE should lie between these two extremes. Finally, to assess the impact of attrition bias on the primary outcomes, we imputed missing outcome data in best-/worst-case and worst-/best-case scenarios [14].

### 2.5. GRADE 

We used GRADE to assess the quality of the body of evidence associated with each outcome [15]. 

## 3. Results

Our search strategy identified 10,374 records. After removal of duplicates and selections based on titles and abstracts, 312 records remained. A total of 271 reports were excluded on the basis of full text, and 41 records reporting 44 RCTs with a total of 22,579 patients were included (Figure 1). 

### 3.1. Characteristics of the Included Studies

Detailed characteristics of the 44 included trials are presented in Appendix A. The year of publication ranged from 1988 to 2018. Forty trials were in English, two in German, one in French, and one in Chinese. Three trials were published as abstracts only, and the Chinese trial was assessed as abstract only due to lacking translation capacity. There were 24 single-center and 20 multicenter trials. Nine different types of LMWH preparations were used, and several types of patients were evaluated: orthopedic or immobilized patients (16 trials), surgical patients (13 trials), ambulatory cancer patients (8 trials), acutely ill medical patients (4 trials), and neurological patients (3 trials). 

### 3.2. Bias Risk Assessment 

Six trials including 8172 patients were considered at overall low risk of bias (Appendix A). Thirty-eight trials were classified as overall high risk of bias. We did not suspect publication bias except for the outcome any VTE, in which asymmetry in the funnel plot was observed (Appendix A). Sensitivity analyses of imputed missing data suggested potential for attrition bias in all primary outcomes, since the imputed effect estimates in the best/worse and worse/best scenario’s suggested benefit and harm, respectively (Appendix A). Results of a post-hoc sensitivity analysis excluding trials published before 2005 were comparable to those of the main analyses (Appendix A).

### 3.3. Co-Primary Outcomes

#### 3.3.1. All-Cause Mortality

Twenty-three trials with 15,487 patients reported data on all-cause mortality, including five trials with 4960 patients at overall low risk of bias. Mortality proportions were 8.0% in the LMWH group and 6.2% in the control group (Figure 2). Meta-analysis of low-risk-of-bias trials showed no statistically significant effect on all-cause mortality (RR 1.03; 95%CI 0.92 to 1.16; *p* = 0.60; *I^2^* = 0%; TSA-adjusted CI 0.88 to 1.20; Table 2). When assessing all trials, the conventional meta-analysis results remained similar, while TSA suggested futility, rejecting a 20% RRR or RRI in mortality. All sensitivity analyses were consistent with the primary analysis (Table 2, Appendix A). Subgroup analyses showed no statistically significant tests of interaction (Appendix A). The overall level of certainty of the evidence was low (Table 2).

#### 3.3.2. Symptomatic Venous Thromboembolism

Twenty-five trials with 15,920 patients reported data on symptomatic VTE, including five trials with 4878 patients at overall low risk of bias. Symptomatic VTE proportions were 1.1% in the LMWH group and 1.8% in the control group (Figure 3 and Figure 4). Meta-analysis of low-risk-of-bias trials showed a statistically significant beneficial effect on symptomatic VTE, which was not confirmed by TSA (RR 0.65; 95%CI 0.45 to 0.94; *p* = 0.02; *I^2^* = 0%; TSA-adjusted CI 0.15 to 3.05; Table 2). When including all trials, both conventional meta-analysis and TSA showed a beneficial intervention effect (RR 0.62; 95%CI 0.48 to 0.81; *p* = 0.0006; *I^2^* = 0%; TSA-adjusted CI 0.44 to 0.89; NNT 137; 97.5%CI 87 to 330; Table 2; Figure 3 and Figure 4). The primary analysis results were confirmed by three out of four sensitivity analyses (Table 2, Appendix A). The direction of the intervention effect consistently suggested benefit in all subgroups, and there were no statistically significant tests of interaction (Appendix A). The overall level of certainty of the evidence was moderate (Table 2).

#### 3.3.3. Major Bleeding

Thirty-three trials with 13,091 patients reported data on major bleeding, including five trials with 4960 patients at overall low risk of bias. Major bleeding proportions were 0.9% in the LMWH group and 0.8% in the control group (Figure 5). Meta-analysis of low-risk-of-bias trials showed a non-statistically significant increase in major bleeding (RR 1.70; 95%CI 0.77 to 3.74; *p* = 0.19; *I^2^* = 0%; Table 2). TSA could not be conducted, since less than 5% of the required information size was accrued. When including all trials, both conventional meta-analysis and TSA showed no statistically significant effect (RR 1.07; 95%CI RR 0.72 to 1.59; *p* = 0.74; *I ^2^*= 0%; TSA-adjusted CI 0.18 to 5.73; Table 2). Sensitivity analyses were consistent with the primary analyses (Table 2, Appendix A). Subgroup analyses showed that low-dose LMWH for more than 30 days was associated with higher risk of major bleeding as compared to shorter treatments (RR 2.20; 95% CI 1.00 to 4.82 vs RR 0.84; 95%CI 0.53 to 1.32, *p* = 0.04 for test of interaction; Appendix A). The overall level of certainty of the evidence was low to moderate (Table 2).

### 3.4. Secondary Outcomes

#### 3.4.1. Serious Adverse Events

Eight trials with 5180 patients reported data on SAE, although events were generally not defined according to ICH-GCP. SAE proportions were 5.4% in the LMWH group and 3.8% in the control group (Appendix A). The one trial at overall low risk of bias, including 1150 patients, showed no statistically significant intervention effect on SAE (RR 0.89; 95%CI 0.68 to 1.17; *p* = 0.42; TSA-adjusted CI 0.41 to 1.96; Table 2). This result was confirmed in both conventional meta-analysis and TSA of all trials regardless of bias risk (RR 0.98; 95% CI 0.78 to 1.25; *p* = 0.89; *I^2^* = 0%; TSA-adjusted CI 0.37 to 2.58; Table 2). As predefined sensitivity analysis, we categorized mortality, symptomatic VTE, and major bleeding events from 37 trials as SAE and used these data to estimate the proportion of patients with one or more SAEs: the results were consistent with those of the primary analysis (Appendix A). Subgroup analyses showed no statistically significant tests of interaction. The overall level of certainty of the evidence was very low to low (Table 2).

#### 3.4.2. Clinically Relevant Non-Major Bleeding

Five trials with 3372 patients reported data on clinically relevant non-major bleeding. Clinically relevant non-major bleeding proportions were 1.0% in the LMWH group and 0.7% in the control group (Appendix A). No trials were at overall low risk of bias. Meta-analysis of all trials showed no statistically significant intervention effect on clinically relevant non-major bleeding (RR 1.50; 95%CI 0.72 to 3.12; *p* = 0.28; *I^2^* = 0%; Table 2), and TSA could not be conducted, since less than 5% of the required information size was accrued. Sensitivity analyses were consistent with the primary analysis (Table 2, Appendix A). Subgroup analyses showed no statistically significant tests of interaction. The overall level of certainty of the evidence was very low (Table 2).

#### 3.4.3. Any Venous Thromboembolism

Thirty trials with 5849 patients reported data on any VTE, including three trials with 1254 patients at overall low risk of bias. Proportions of any VTE were 10.7% in the LMWH group and 17.6% in the control group (Appendix A). Meta-analysis of the low risk of bias trials showed a statistically significant beneficial effect on any VTE, which was not confirmed by TSA (RR 0.57; 95%CI 0.38 to 0.84; *p* = 0.005; *I^2^* = 0%; TSA-adjusted CI 0.11 to 2.82; Table 2). When including all trials, both conventional meta-analysis and TSA showed a beneficial intervention effect (RR 0.61; 95%CI 0.50 to 0.75; *p* < 0.00001; *I^2^* = 47%; TSA-adjusted CI 0.49 to 0.82; NNT 15; 97.5%CI 11 to 21; Table 2). The primary analysis results were confirmed by all sensitivity analyses (Table 2, Appendix A). Subgroup analyses showed no statistically significant tests of interaction. The overall level of certainty of the evidence was low to moderate (Table 2).

## 4. Discussion

In this systematic review on low-dose LMWH versus placebo or no treatment, LMWH was not associated with a statistically significant intervention effect on mortality, major bleeding, clinically relevant non-major bleeding, or SAE. Conversely, we found a large beneficial intervention effect on both symptomatic VTE and on any VTE which included asymptomatic events detected through screening. These effects were consistent among subgroup and sensitivity analyses, but the effect size varied per patient type, and the quality of the evidence was moderate. In the TSAs of mortality, symptomatic VTE, and any VTE, the adjusted monitoring boundaries were crossed (respectively, for futility and for benefit), indicating a low risk of random error. The intervention effects of low-dose LMWH on SAE and bleeding events remain inconclusive, as TSA monitoring boundaries were not crossed, and quality of evidence was low. There was a suggestion of publication bias in the reporting of any VTE, and attrition bias may have influenced the primary outcomes. 

### 4.1. Considerations on the Optimal Prophylactic Dose 

Previous systematic reviews did not observe a mortality benefit for patients receiving LMWH thrombosis prophylaxis compared to patients receiving placebo or no treatment, which is confirmed by our results including TSA. Although it was previously thought that LMWHs might improve survival in cancer patients, later systematic reviews found no survival benefit in cancer patients receiving different prophylactic doses of LMWH [6,9]. Additionally, we detected no beneficial effect on mortality in any patient category in a previous meta-analysis on intermediate-dose LMWH [11]. Nevertheless, we cannot exclude the possibility of a smaller intervention effect than 20% RRR/RRI on mortality; this would require many more randomized patients, as we used a 20% RRR for calculating the required information size in TSA. 

In line with previous literature, we found a consistent beneficial intervention effect on VTE in subgroup analyses according to patient type, although effect sizes varied among subgroups. The overall incidence of symptomatic VTE was low, resulting in an NNT of 137. Effect estimates were rather similar regardless of bias risk (low risk RCTs estimated an RRR of 35%, while all RCTs combined estimated an RRR of 41%), suggesting we may base our conclusions on the more accurate estimates derived from the meta-analyses of all trials. Previous systematic reviews on thrombosis prophylaxis have found larger relative risk reductions [6,7,10,25]. This could indicate that low-dose LMWH may be slightly less effective for the prevention of VTE than more frequently used higher doses. However, this indirect comparison should be viewed with caution, as differences between reviews regarding study selection criteria could also explain the difference. A direct comparison in a homogeneous patient population is required for strong inferences about the efficacy of low-dose LMWH compared to higher doses.

Finally, evidence on adverse events remains inconclusive. The point-estimate of the low-risk-of-bias trials suggested a 70% RRI in major bleeding which was not statistically significant, while the estimate including all trials was neutral. This difference may relate to bias risk but could also be explained by trial characteristics: cancer and treatment duration are risk factors for major bleeding, and three out of five low-risk-of-bias trials included oncological patients who were generally treated for a longer duration [26]. The increased risk of major bleeding in the subgroup of oncological patients was comparable to that reported in previous systematic reviews for this patient category [6,7]. Conversely, the risk of major bleeding for other patient types was low compared to that indicated in other systematic reviews [10,11,25]. This may be explained by the low LMWH dose but also by differences in included patients or co-interventions. Data on clinically relevant non-major bleeding were reported by only a few trials, and analyses were inconclusive. There was no apparent effect on SAE, confirmed by sensitivity analyses in which we incorporated data from nearly all available trials. Assessment of these two outcomes was hampered by wide variations in definitions and reporting between trials, resulting in low- to very low quality evidence and limiting inferences on the harms of low-dose LMWH.

### 4.2. Implications for Clinical Practice

In general, clinicians will not prescribe thrombosis prophylaxis without considering both effectiveness and harms. This balance may differ depending on patients’ characteristics such as disease type, severity of illness, or surgery. In prespecified subgroup analyses according to patient type, we found that, in surgical patients, low-dose LMWH reduced both symptomatic and any VTE, without evidence for increased major bleeding. In orthopedic patients, there was a statistically significant reduction in any, but not in symptomatic, VTE, with no evidence for increased major bleeding events. Although not statistically significant, there was a 39% RRR in symptomatic VTE, and the discrepancy may be explained by low power. In oncological patients, a beneficial effect on symptomatic VTE, but not on any VTE, was found. Additionally, the direction of the intervention effect suggested an increase in major bleeding. There were no statistically significant beneficial or harmful effects in acutely ill medical patients, suggesting either that there was a very small intervention effect with concurrent high numbers needed to treat or that a low LMWH dose is insufficient for this type of patient. Recent guidelines have recommended an individualized approach towards thrombosis prophylaxis in acutely ill medical patients [2]. On the basis of our results, one could hypothesize that medical patients deemed at high risk of VTE will mainly benefit from higher doses of thrombosis prophylaxis. Finally, only very few neurological patients were included, limiting inferences for this subgroup.

This systematic review provides a general overview of the effects of low-dose LMWH: although there are differences between patient subgroups, there also are many similarities in the direction of effects. Overall, we found that low-dose LMWH was most effective in surgical, orthopedic, and oncological patients, while the estimated RRI for bleeding events was low in most prespecified patient subgroups, except for oncological patients. These results should be viewed in the perspective of the limited quality of the evidence and the inherent limited power of subgroup analyses. In cases where physicians are in doubt whether a patient should receive thrombosis prophylaxis or no prophylaxis at all or when a higher prophylactic dose is deemed inappropriate with respect to bleeding risk, clinicians may consider a low-dose LMWH for thrombosis prophylaxis, especially in surgical and orthopedic patients. 

### 4.3. Strengths and Limitations 

Strengths of this review include its systematic and transparent methodology according to recommendations by the Cochrane Handbook, the PRISMA statement, and the GRADE working group. We used a prespecified protocol, a comprehensive search strategy without language restrictions, although we did assess one Chinese article as abstract only, independent data extraction and bias assessment by two authors, and incorporation of bias risk assessment in the results and conclusions. Finally, we applied TSA to all outcomes to assess the risks of random error and to estimate the required information size.

Nevertheless, several important limitations apply. Our main goal was to make general inferences on the efficacy and safety of low-dose LMWH, using all available evidence. Consequently, there was a high amount of clinical heterogeneity between trials. The balance between thrombosis and bleeding may vary depending on patient subgroup characteristics: relying on overall effect estimates could obscure more subtle associations or lead to wrong inferences about a subpopulation. However, the distinction between different patient populations is somewhat arbitrary in any systematic review, and we attempted to account for clinical heterogeneity by conducting several preplanned subgroup analyses. This approach offers the benefit of increased power of the meta-analysis, and we found the direction of the intervention effects was equal in most subgroups.

A second limitation concerns the inclusion of trials comparing low-dose LMWH to an inactive comparator, which led to the selection of mainly older trials or trials assessing LMWH in specific patient types or countries, limiting the generalizability of our results. In a post-hoc sensitivity analysis excluding trials published before 2005, the results remained comparable, although no inferences could be made for subgroups due to the very limited sample size. 

Third, to estimate the effect of low-dose LMWH on SAEs we conducted a sensitivity analysis to estimate the proportion of patients having one or more SAEs. For this purpose, we categorized mortality, symptomatic VTE, SAE, and major bleeding events from 37 trials as SAE. In reality, not all symptomatic VTE and major bleeding events are SAEs by definition (i.e., a distal leg thrombosis may be classified as adverse event, while pulmonary embolism can be a serious adverse event), but making this distinction was impossible on the basis of insufficiently detailed trial reports. Last, the best-/worst- and worst-/best-case analyses we performed to explore the influence of missing outcome data were probably overpowered to detect potential attrition bias, since the incidence of lost to follow-up was higher than the incidence of the primary outcomes. 

## 5. Conclusions

In a wide variety of patients at risk of VTE, there was very low to moderate-quality evidence that low-dose LMWH for thrombosis prophylaxis did not decrease all-cause mortality but reduced the incidence of symptomatic and asymptomatic VTE, while results on the intervention effects on major bleeding, clinically relevant non-major bleeding, and SAE remain inconclusive. 

## Figures and Tables

**Figure 1 jcm-08-02039-f001:**
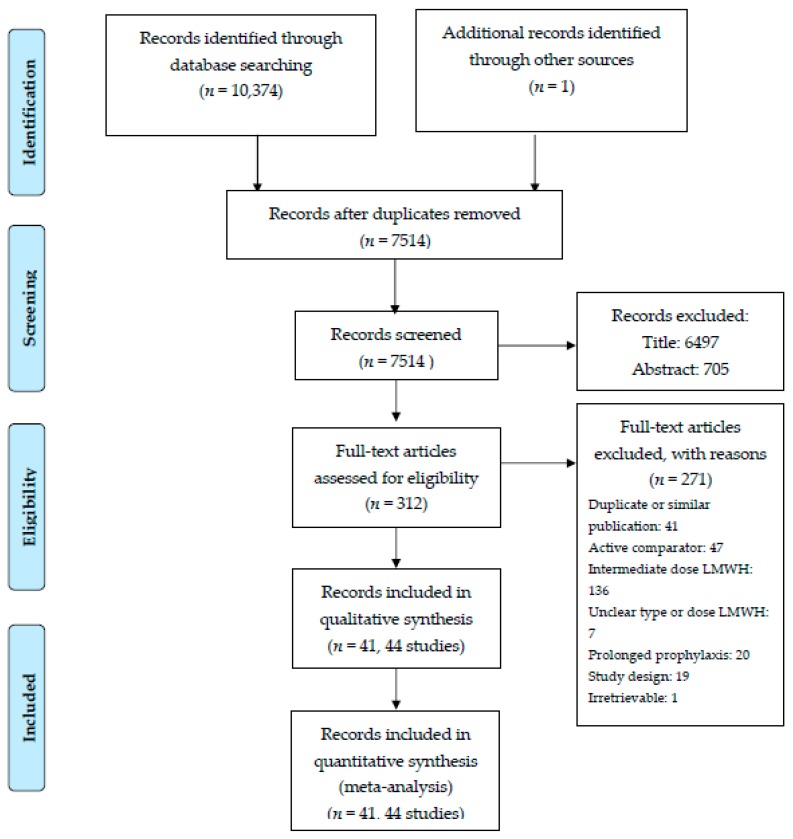
Preferred reporting items for systematic reviews and meta-analyses (PRISMA) flow-chart of study inclusion.

**Figure 2 jcm-08-02039-f002:**
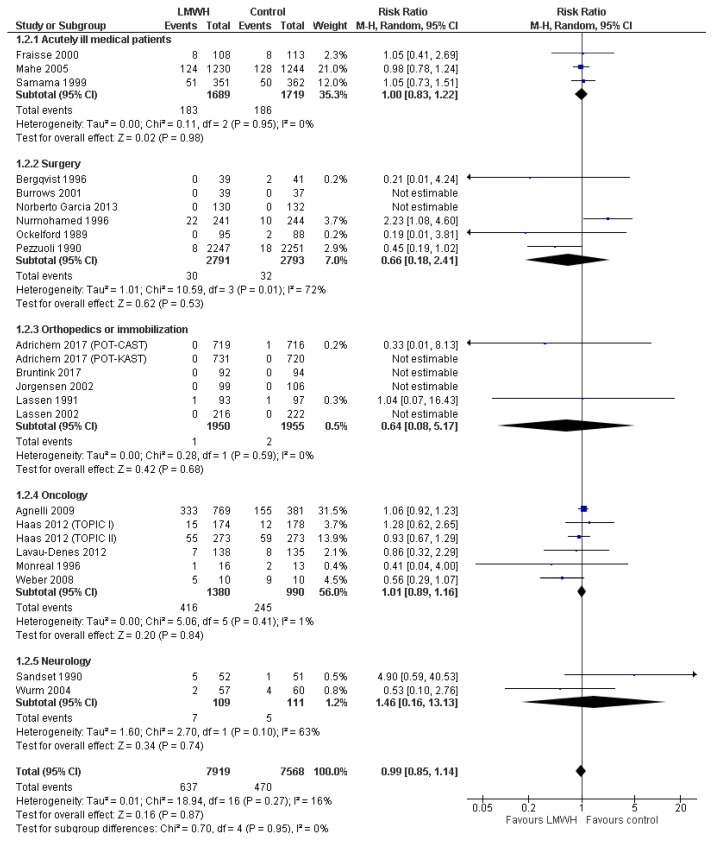
Forest plot of all-cause mortality. Forest plot of all-cause mortality at maximal follow-up of LMWH prophylaxis compared to placebo or no treatment, stratified according to population. The size of the squares reflects the weight of the trial in the pooled analysis. Horizontal bars represent 95% confidence intervals; LMWH, low-molecular-weight heparin; CI, confidence intervals.

**Figure 3 jcm-08-02039-f003:**
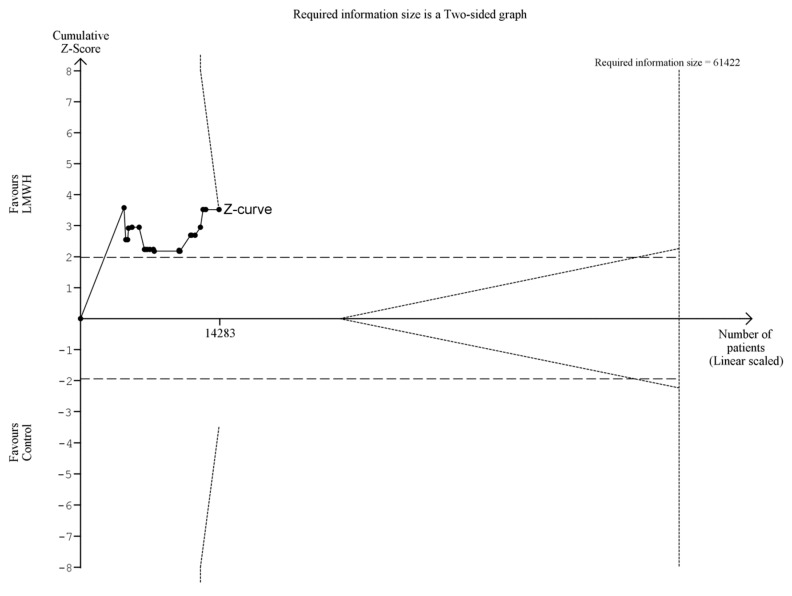
Trial sequential analysis of symptomatic venous thromboembolism (VTE). Trial sequential analysis of symptomatic VTE at maximal follow-up of LMWH compared to placebo or no treatment. The required information size was calculated using α = 0.025, β = 0.90, relative risk reduction (RRR) = 20%, diversity (D2) as suggested by trials, and a control event rate of 1.81%. The cumulative Z-curve was constructed using a random-effects model, and each cumulative Z-value was calculated after inclusion of a new trial (represented by black dots). The dotted horizontal lines represent the conventional naïve boundaries for benefit. The etched lines represent the trial sequential boundaries for benefit (positive), harm (negative), or futility (middle triangular area). The cumulative Z-curve crosses the TSA boundary for benefit, indicating future trials are very unlikely to change the conclusions. Note: the two most recent trials were excluded from this TSA because inclusion would result in an incorrect graphical display of the LanDeMets boundary for benefit. The TSA-adjusted confidence interval remained similar.

**Figure 4 jcm-08-02039-f004:**
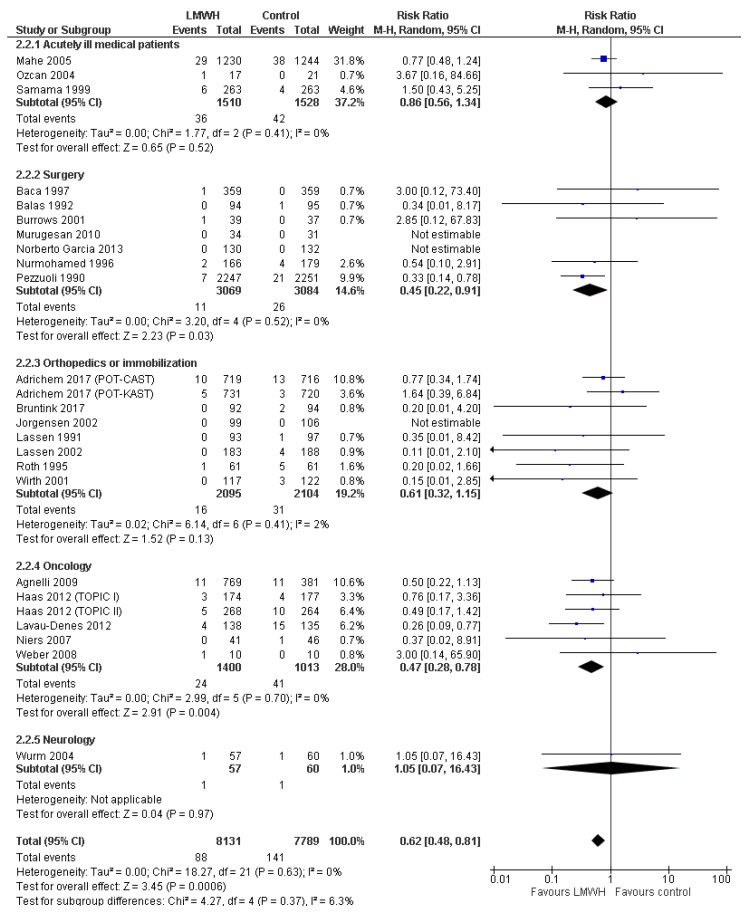
Forest plot of symptomatic VTE. Forest plot of symptomatic VTE at maximal follow-up of LMWH prophylaxis compared to placebo or no treatment, stratified according to patient type. The size of the squares reflects the weight of the trial in the pooled analysis. Horizontal bars represent 95% confidence intervals.

**Figure 5 jcm-08-02039-f005:**
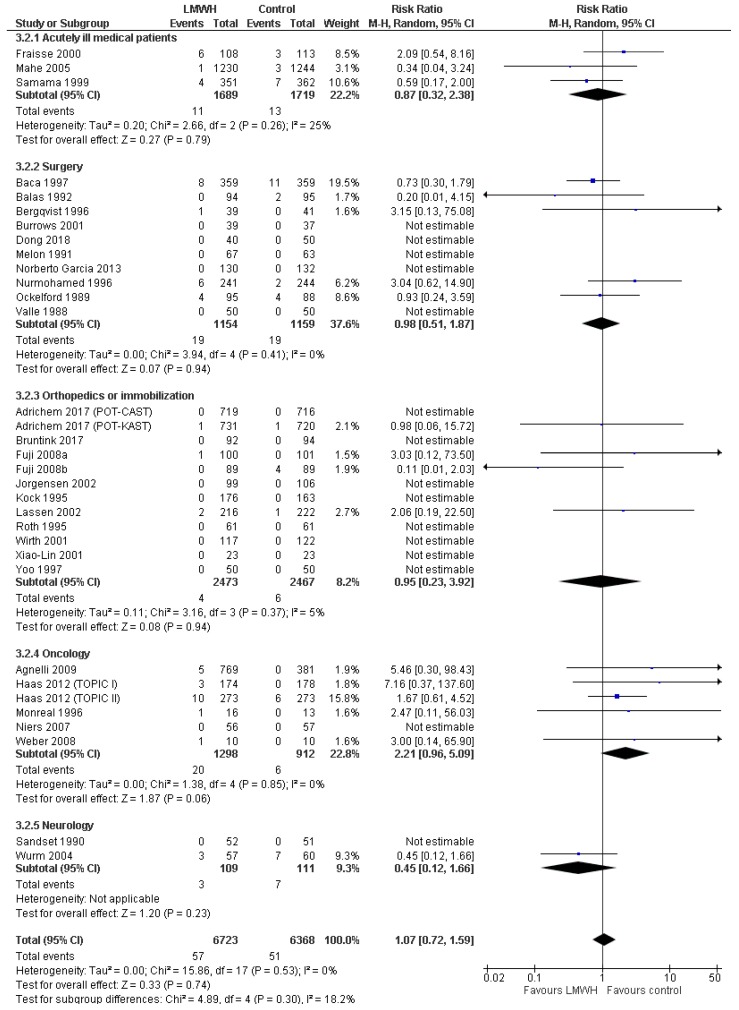
Forest plot of major bleeding. Forest plot of major bleeding at maximal follow-up of LMWH prophylaxis compared to placebo or no treatment, stratified according to patient type. The size of the squares reflects the weight of the trial in the pooled analysis. Horizontal bars represent 95% confidence intervals.

**Table 1 jcm-08-02039-t001:** LMWH dose definitions.

LMWH Type	A Priori Defined as Low-Dose LMWH	Dose Used in Included Trials
Bemiparin	<3500 IU	2500 IU
Certoparin	<5000 IU	3000 IU
Dalteparin	<5000 IU	2500 IU ^a^
Enoxaparin	<40 mg	20 mg
Nadroparin	<5700 IU	2850–3800 IU ^b,c^
Parnaparin	<4250 IU	3200 IU
Reviparin	<3436 IU	1750 IU
Tinzaparin	<4500 IU	3500 IU ^d^

IU: International Units; LMWH: low-molecular-weight heparin; mg: milligrams; ^a^ Sandset et al. used weight-dependent doses of 3000–5500 IU; ^b^ Fraisse et al. used weight-dependent doses of 3800–5700 IU; ^b^ Yoo et al. used weight-dependent doses of 2850–5700 IU; ^c^ Xiao-Li et al. used weight-dependent doses of 41–62 IU/kg; ^d^ Sorensen et al. and Lassen et al. used weight-dependent doses: 50 IU/kg.

**Table 2 jcm-08-02039-t002:** Conventional meta-analysis and trial sequential analysis outcomes.

Outcome	Included Trials	Trials (Patients)	Conventional Meta-Analysis ^a^	Primary TSA ^a^ α 2.5%; ß 90%; RRR 20%; D^2^ Model Variance Based	Sensitivity TSA ^a^ α 2.5%; ß 90%; RRR Based on Low Risk Trials; D^2^ Model Variance Based	Sensitivity TSA ^a^ α 2.5%; ß 90%; RRR 20%; D^2^ 25%	Certainty of Evidence
Mortality	Low bias risk	5 (4.960)	RR 1.03 (0.92 to 1.16)	RR 1.03 (0.88 to 1.20)	Insufficient data (<5% of DIS)	RR 1.03 (0.86 to 1.23)	Low ^d, e, f^
	All	23 (15.487)	RR 0.99 (0.85 to 1.14)	RR 1.02 (0.89 to 1.16) ^b^	Insufficient data (<5% of DIS)	RR 1.02 (0.90 to 1.15)	Low ^d, g^
Symptomatic VTE	Low bias risk	5 (4.878)	RR 0.65 (0.45 to 0.94)	RR 0.67 (0.15 to 3.05)	RR 0.67 (0.32 to 1.38)	0.67 (0.15 to 3.05)	Moderate ^e^
	All	25 (15.920)	RR 0.62 (0.48 to 0.81)	RR 0.62 (0.44 to 0.89) ^c^	RR 0.62 (0.42 to 0.92)	RR 0.62 (0.20 to 1.95)	Moderate ^g^
Major bleeding	Low bias risk	5 (4.960)	RR 1.70 (0.77 to 3.74)	Insufficient data (<5% of DIS)	Insufficient data (<5% of DIS)	Insufficient data (<5% of DIS)	Moderate ^f^
	All	33 (13.091)	RR 1.07 (0.72 to 1.59)	RR 1.01 (0.18 to 5.73) ^c^	RR 1.01 (0.52 to 1.93)	RR 1.09 (0.75 to 1.60)	Low ^e, f, g^
SAE	Low bias risk	1 (1.150)	RR 0.89 (0.68 to 1.17)	RR 0.89 (0.41 to 1.96)	RR 0.89 (0.27 to 2.96)	RR 0.89 (0.36 to 2.23)	Low ^e, f, h^
	All	8 (5.180)	RR 0.98 (0.78 to 1.25)	RR 0.98 (0.37 to 2.58)	Insufficient data (<5% of DIS)	RR 0.98 (0.77 to 1.24)	Very low ^d, e, f, g^
Clinically relevant non-major bleeding	Low bias risk	0 (0)	-	-	-	-	-
	All	5 (3.372)	RR 1.50 (0.72 to 3.12)	Insufficient data (<5% of DIS)	Insufficient data (<5% of DIS)	Insufficient data (<5% of DIS)	Very low ^d, e, f, g^
Any VTE	Low bias risk	3 (1.254)	RR 0.57 (0.38 to 0.84)	RR 0.57 (0.11 to 2.82)	RR 0.57 (0.32 to 1.01)	Not performed (D^2^ >25%)	Moderate ^e, i, k^
	All	30 (5.849)	RR 0.61 (0.50 to 0.75)	RR 0.63 (0.49 to 0.82)	RR 0.63 (0.50 to 0.80)	Not performed (D^2^ >25%)	Low ^e, i, j, k^

^a^ Small discrepancies of the intervention effect estimates between traditional RevMan meta-analyses and the TSA-adjusted results may occur due to different pooling methods (for example the inclusion of zero-event trials in TSA analyses); ^b^ TSA monitoring boundary for futility crossed; ^c^ sensitivity analysis using Peto’s odds ratio showed similar results; ^d^ downgraded for inconsistency, since point estimates varied widely; ^e^ downgraded for imprecision, since TSA-adjusted confidence interval crossed ‘1’; ^f^ downgraded for imprecision, since conventional confidence interval crossed ‘1’; ^g^ downgraded for risk of bias, since (some) included trials were at high risk of bias; ^h^ downgraded for indirectness, since only one trial was included under assessment; ^i^ downgraded for risk of publication bias, since there was important asymmetry in the funnel plot; ^j^ downgraded for inconsistency, since point estimates varied widely and there was moderate statistical heterogeneity; ^k^ upgraded, since there was a strong association. α: two-sided significance level, ß: power; D^2^: diversity; DIS: diversity-adjusted information size; OR: odds ratio; RR: relative risk; RRR: relative risk reduction; SAE: serious adverse events; TSA: trial sequential analysis; VTE: venous thromboembolism.

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
