# Peer review of "Low Dose Low-Molecular-Weight Heparin for Thrombosis Prophylaxis: Systematic Review with Meta-Analysis and Trial Sequential Analysis"

_jcm, 2019, doi:10.3390/jcm8122039_

Round 1
Reviewer 1 Report
The authors have addressed all my concerns effectively in this revision.
Author Response
We thank the reviewer for his/her comments, and we are happy that we were able to address these effectively.
Reviewer 2 Report
Thank you for addressing previous concerns. I think the addition of the forest plots into the main text make this article much easier to digest.
Author Response
We thank the reviewer for his or her comments, and we are happy that we were able to address these effectively.
This manuscript is a resubmission of an earlier submission. The following is a list of the peer review reports and author responses from that submission.
Round 1
Reviewer 1 Report
It is clear that every effort was made to employ an organized and methodical approach to analyzing the heterogeneous data.
Because there is so much data to absorb, I would recommend including the forest plots in Supplementary Figures S1-S5 in the main text. These forest plots are a nice way to visually absorb the information described in the text, which can get a bit wordy otherwise.
I'm not sure I understand the best/worst case and worst/best case analysis and because it is an unfamiliar analysis to me, I'm not sure it adds much to the overall conclusions, other than to further support a thorough analytical approach.
I would also recommend including Supplementary Table S4 in the main text. When I think of "low dose LMWH", I think of prophylactic doses of enoxaparin (e.g. 40mg once daily) or dalteparin (e.g. 5000 units once daily) and I did not realize until I looked through the supplementary material that the trials included even lower doses than this.
Reviewer 2 Report
In general, this is a well-done systematic review and meta-analysis on always an important clinical issue.
Authors' conclusions should be somewhat cautioned and toned down in the light of their findings, there are several issues:
Most of the evidence regarding proposed conclusions have been derived on the low-quality to medium quality studies at best, with the certainty of evidence ranging from very low to moderate. This should be emphasized throughout the text and also amended in the author's conclusion. Second issue is that LMWH administration seemed to reduce symptomatic VTE in the setting of surgical and oncology patients while this effect was absent in the setting of acutely ill, orthopedics or neurology patients, when stratified by pre-specified groups - this should also be reworded and emphasized in the text, it seems that targeting these groups showed the highest benefit with low-molecular heparin. On the other hand, any VTE prevention was significant and beneficial in orthopedics and surgery patients, but not in neurology, acute medical illness or oncology patients. The authors should comment on these discrepancies and offer a possible explanation as to why was this observed. I am not sure what this study adds to the literature. We know from this study that LMWH administration is better than no LMWH administration in terms of symptomatic VTE and some other endpoints, but we cannot ascertain bleeding risks associated with this dose, therefore we cannot truly estimate cost-benefit ratio in this setting due to the low quality of evidence. Secondly, it would be more informative to know how does a low-dose LMWH fare in comparison to medium-dose LMWH or high dose LMWH in terms of the outcomes than when it is compared to a placebo alone. I believe the former would be much more practice-informing. LMWH administration did not affect all-cause mortality suggesting that the vast majority of deaths in these heterogeneous groups of disorders had other underlying culprits of deaths rather than thromboembolic cause Some of the more robust studies adding a lot to study weight were quite old, some of them even 30 years old which makes possible interpretation of results quite difficult and perhaps impossible to translate to today's modern clinical practice given the relevant changes in baseline characteristics of population, improved modalities of treatment and pharmacology, it would be interesting to see how these outcomes would change if only newer studies were included, e.g. in last 10 or 15 years I would wish to commend authors for their effort and for a study being methodologically very well-performed and pre-specified, however, authors should provide more evidence what these results mean for daily clinical practice and what subgroups are most likely to benefit from a low dose of LMWH prophylaxis and, in contrast, which would not. At all times, pertinent limitations should be kept at bay and cautiously acknowledged.